# Urine metabolomic profiles of autism and autistic traits–A twin study

**Abishek Arora**[1,2], **Francesca Mastropasqua**[1,2], **Sven Bölte**[1,3,4],
**Kristiina Tammimies**[1,2]*

1 Department of Women's and Children's Health, Center of Neurodevelopmental Disorders (KIND), Centre for Psychiatry Research, Karolinska Institutet, Stockholm, Sweden, 2 Astrid Lindgren Children's Hospital, Karolinska University Hospital, Region Stockholm, Stockholm, Sweden, 3 Child and Adolescent Psychiatry, Stockholm Health Care Services, Region Stockholm, Stockholm, Sweden, 4 Curtin Autism Research Group, Curtin School of Allied Health, Curtin University, Perth, Western Australia

* kristiina.tammimies@ki.se

**Data Availability Statement:** The data is accessible, after necessary clearances, through the Swedish National Data Service's (SND) research data catalogue and stored at the KI Data Repository (KI DR) (doi:available after acceptance). The

## Abstract

Currently, there are no reliable biomarkers for autism diagnosis. The heterogeneity of autism and several co-occurring conditions are key challenges to establishing these. Here, we used untargeted mass spectrometry-based urine metabolomics to investigate metabolic differences for autism diagnosis and autistic traits in a well-characterized twin cohort (N = 105). We identified 208 metabolites in the urine samples of the twins. No clear, significant metabolic drivers for autism diagnosis were detected when controlling for other neurodevelopmental conditions. However, we identified nominally significant changes for several metabolites. For instance, phenylpyruvate (p = 0.019) and taurine (p = 0.032) were elevated in the autism group, while carnitine (p = 0.047) was reduced. We furthermore accounted for the shared factors, such as genetics within the twin pairs, and report additional metabolite differences. Based on the nominally significant metabolites for autism diagnosis, the arginine and proline metabolism pathway (p = 0.024) was enriched. We also investigated the association between quantitative autistic traits, as measured by the Social Responsiveness Scale 2nd Edition, and metabolite differences, identifying a greater number of nominally significant metabolites and pathways. A significant positive association between indole-3-acetate and autistic traits was observed within the twin pairs (adjusted p = 0.031). The utility of urine biomarkers in autism, therefore, remains unclear, with mixed findings from different study populations.

## Introduction

Autism is a neurodevelopmental condition with a heterogeneous presentation that is diagnosed in 1 to 2% of the world population [1]. Although several genes and genetic variants have been associated with autism phenotypes [1–3], there is no evidence of a single biomarker that can be of benefit to the diagnostic procedure of the condition [4]. Finding biomarkers that could aid either in the diagnosis or specifying a subgroup of individuals could help, for instance, in the early diagnosis of autism [5].

utilised code is available on GitHub (https://github.com/Tammimies-Lab/RATSS-Metabolomics) or available upon request from the corresponding author."

**Funding:** The project was supported by the Swedish Research Council (S.B., and K.T.), Swedish Foundation for Strategic Research (K.T.), the Swedish Brain Foundation – Hjärnfonden (K.T.), the Harald and Greta Jeanssons Foundations (K.T.), Åke Wiberg Foundation (K.T.), Strategic Research Area Neuroscience Stratneuro (K.T.), The Swedish Foundation for International Cooperation in Research and Higher Education STINT (K.T.), and Board of Research at Karolinska Institutet (K.T.). The funders had no role in study design, data collection and analysis, decision to publish, or preparation of the manuscript.

**Competing interests:** A.A., F.M., and K.T. declare no competing interests. S.B. declares no direct conflict of interest related to this article. He discloses that he has in the last three years acted as an author, consultant, or lecturer for Medice and Roche. He receives royalties for textbooks and diagnostic tools from Hogrefe, Kohlhammer, UTB and Liber. S.B. is a partner in NeuroSupportSolutions International AB.

Prior research has suggested an association of metabolic anomalies with autism that could potentially aid in the search for biomarkers [6]. Metabolomic analyses using urine samples provide a non-invasive method with greater ease of sample collection and handling in comparison to other biospecimens. This is particularly advantageous in clinical settings where the process of diagnosing autism can begin at a younger age [7]. Moreso, the ability to detect metabolites in urine samples is well-established using mass spectrometry (MS) and nuclear magnetic resonance spectroscopy (NMR), and the typical human urine metabolome has been mapped [8]. Early efforts to understand the urine metabolome in autism began several decades ago, with initial biochemical studies searching for associations [9, 10]. The investigations have shifted to more untargeted and high-precision analytical approaches using NMR [11, 12] and MS [13–15].

The interest in autism metabolomics has continued to expand; however, more accessible data are required across different populations to identify robust associations. Such an approach will also make it possible to characterise how both, situational and long-term environmental factors interplay with the genetic backgrounds of autism [16]. Twin studies are an appropriate tool to investigate the contribution of environmental and genetic factors [17, 18] and can be extended to metabolic profiles. It is especially valuable to analyse differences between monozygotic (MZ) twins who are discordant for autism, to tease out genetic contributions.

In this study, we utilised a rare twin sample, the Roots of Autism and ADHD Twin Study in Sweden (RATSS) [19, 20], enriched for MZ twins discordant for autism, and their urine sample-based metabolomics to search and validate potential metabolite biomarkers. Furthermore, on the basis of these findings we have mapped the affected metabolic pathways. Our study adds on to the literature exploring the metabolites as possible biomarkers for autism and expands the search for autistic traits.

## Materials and methods

### Study participants

In this study, individuals (N = 105) and 48 complete twin pairs were selected from the RATSS cohort [19, 20], which is a neurodevelopmental condition (NDC) enriched twin sample recruited from the general Swedish population between June 2011 and December 2015, for untargeted mass spectrometry-based urine metabolomics. Detailed inclusion and exclusion criteria for RATSS have been previously published [19]. Among the recruited twins, we selected participants for this study based on the autism diagnosis status, whether the twin pair was concordant (both with autism diagnosis) or discordant (only one with autism diagnosis) and if they had available urine samples. Furthermore, we age- and sex-matched the non-autistic twin pairs. The study was approved by the Swedish Ethical Review Authority (2016/1452-31). Written informed consent was obtained from all participants or their caregivers, based on their age.

During a 2.5-day study visit, a team of clinical professionals conducted a diagnostic evaluation of the participants in line with the Diagnostic and Statistical Manual of Mental Disorders, 5th Edition (DSM-5) guidelines [21]. The evaluation utilised a combination of diagnostic interviews, a review of medical history documents, and the use of established diagnostic measures [19]. This included behavioural assessment tools such as the Autism Diagnostic Interview-Revised (ADI-R) [22], the Autism Diagnostic Observation Schedule 2nd Edition (ADOS-2) [23]. Furthermore, additional tools were used to establish the diagnosis of other NDCs, if any, such the Diagnostic Interview for ADHD in Adults (DIVA-2) [24], the Structured Clinical Interview for DSM-IV (SCID-IV) [25], and the Adaptive Behavior Assessment

System (ABAS) [26]–more detailed information about the same is available in the publication by Bölte and colleagues [19]. Autistic traits were evaluated with the parent-report version of the Social Responsiveness Scale 2nd Edition (SRS-2) [27–29], consisting of 65 items. Intelligence quotient (IQ) was measured by using the Wechsler Intelligence Scale for Children or Adults—IV General Ability Index (GAI) [30].

Additionally, the participants were asked for a list of their current, regularly used medications during the study visit. As there was a collection of different medications including antidepressants and ADHD medication, all were grouped together and adjusted for in our analyses. No subgrouping was possible for the drugs due to lack of power to detect metabolomic effects of specific drugs.

## Urine collection and metabolite extraction

The urine samples were collected at the last day of the visit from the study participants. First, the participants were informed about the urine sample collection into a special urine cup. The urine cup was then given to a research nurse who transferred 10 mL of the collected urine to a sterile vacutainer tube with no additives. The sample was then directly transported, aliquoted and stored in the Karolinska Institutet Biobank at -80˚C. The collected samples were further transported for analysis to the Proteomics and Metabolomics Facility, University of Tuscia, Italy. All samples were handled as per the same stated protocol. Before the metabolomic analysis, the urinary specific gravity was measured following centrifugation at 13,000 g for 10 minutes. Urine aliquots (200 μl) were mixed with 200 μl of methanol:acetonitrile:water (50:30:20), vortexed for 30 minutes, maximum speed at 4˚C and then centrifuged at 16,000 g for 15 minutes at 4˚C. Supernatants were collected for metabolomic analysis.

## Ultra-High Performance Liquid Chromatography (UHPLC)

For the experiments, 20 μL of samples were injected into a UPLC system (Ultimate 3000, Thermo Scientific) and were analysed on positive mode: samples were loaded onto a Reprosil C18 column (2.0 mm × 150 mm, 2.5 μm—Dr Maisch, Germany) for metabolite separation. Chromatographic separations were achieved at a column temperature of 30˚C and flow rate of 0.2 mL/min. A linear gradient (0–100%) of solvent A (ddH$_2$O, 0.1% formic acid) to B (acetonitrile, 0.1% formic acid) was employed over 20 minutes, returning to 100% solvent A in 2 minutes and a 6-minute post-time solvent A hold. Acetonitrile, formic acid, and HPLC-grade water were purchased from Sigma Aldrich.

## High Resolution Mass Spectrometry (HRMS)

The UPLC system was coupled online with a mass spectrometer, Q Exactive (Thermo Scientific), scanning in full MS mode (2 μ scans) at a resolution of 70,000 in the 67 to 1000 $m/z$ range, target of $1 \times 106$ ions and a maximum ion injection time (IT) of 35 ms, 3.8 kV spray voltage, 40 sheath gas, and 25 auxiliary gas, operated in negative and then positive ion mode. Source ionization parameters were: spray voltage, 3.8 kV; capillary temperature, 300˚C; and S-Lens level, 45. Calibration was performed before each analysis against positive or negative ion mode calibration mixes (Piercenet, Thermo Fisher, Rockford, IL) to ensure sub-ppm error of the intact mass.

## Metabolite quantification

Data were normalized by urinary specific gravity because creatinine excretion may be abnormally reduced in autistic children [31]. Replicates were exported as mzXML files and

processed through MAVEN [32]. Mass spectrometry chromatograms were elaborated for peak alignment, matching and comparison of parent and fragment ions, and tentative metabolite identification (within a 10-ppm mass deviation range between observed and expected results against the imported Kyoto Encyclopaedia of Genes and Genomes (KEGG) database [33].

## Differential metabolomics

Principal component analysis (PCA) was performed in R [34] followed by visualisation with the *factoextra* package (https://cran.r-project.org/package=factoextra) to identify any outliers in the dataset. Prior to the PCA, missing values (0.018%) were imputed using the *missMDA* package [35] in R [34]. Here, the number of dimensions of the PCA were estimated by cross-validation using the k-fold method followed by imputation of missing values using the PCA model.

Differential metabolomic analysis for 208 metabolites was performed using the *drgee* package [36] in R [34], for overall effects and differences between twin pairs using a generalised estimating equations (GEE) model, with suitable covariates. For this analysis, missing values (0.018%) were not removed from the dataset and were treated as *NA* values. In the GEE model, the metabolites were individually considered as the response variable, while the autism diagnosis status was the predictor variable. Age, sex, body mass index (BMI), medication status and diagnosis status of other NDCs were added as covariates to the model: *Metabolite ~ Autism Diagnosis + Other NDC Diagnosis + age + sex + BMI + medication status*. The model was applied for the whole cohort (Model A) and for differences within twin pairs (Model B) through the *drgee* package [36] in R [34]. The same model was then applied to a subset of the cohort based on zygosity. Here, age and sex were removed as covariates when testing for differences between twin pairs as these were intrinsically controlled.

In addition to testing autism diagnosis status as a predictor variable, we used autistic traits measured by the SRS-2 [27–29] total raw scores. As IQ levels are known to have an impact on the SRS-2 scores [37], along with the previously stated covariates (other than diagnostic status of other NDCs), IQ scores from GAI were added to the models: *Metabolite ~ SRS total raw score + age + sex + BMI + medication status + IQ GAI*. Also, for SRS-2 total raw scores, the models were applied to subsets of the cohort based on zygosity. As previously stated, age and sex were removed as covariates when testing for differences between twin pairs as these were intrinsically controlled. For all instances, p values were extracted from the results of each model and adjusted for multiple comparisons using the false discovery rate (FDR) method [38] in R [34].

## Pathway enrichment analysis

Significant metabolites from the GEE models were tested for enrichment in the Small Molecule Pathway Database (SMPDB) [39] using over-representation analysis (ORA) from the enrichment analysis module of MetaboAnalyst 5.0 [40]. The cut-off threshold for entries from metabolite sets to be included was set to 2. ORA was implemented using the hypergeometric test to evaluate whether a particular metabolite set is represented more than expected by chance within the given list of metabolites. One-tailed p values were provided after adjusting for multiple testing. Based on the statistical testing, enrichment dot-plots and pathway networks were generated. In the pathway networks, 2 or more nodes were only connected by an edge when the number of shared metabolites was >25% of the combined metabolites contributing to each node.

## Statistical analysis and plots

All statistical analyses were performed in R (v4.1.2) [34]. The statistical models and tests used for the analyses are described in the methodology relevant to the experimental technique, in the sections above. All plots, unless otherwise stated, were created using the *ggplot2* package (v3.3.5) [41] or the *pheatmap* package (v1.0.12, https://CRAN.R-project.org/package= pheatmap) in R (v4.1.2) [34].

## Results

### Metabolites do not clearly explain sample differences

UHPLC was coupled with HRMS for the untargeted analysis of metabolites. In the study cohort (N = 105 twins), of which 48 form complete twin pairs, we identified 208 metabolites with high reliability following mass spectrometry of urine samples. Principal component analysis (PCA) of the metabolomics data demonstrated no clear drivers of effects across PC1 (23.2%) and PC2 (10.8%) (Fig 1A). The first 10 principal components were able to explain 68.2% of the variation (Fig 1B). The loadings for PC1 (Fig 1C) and PC2 (Fig 1D) did not

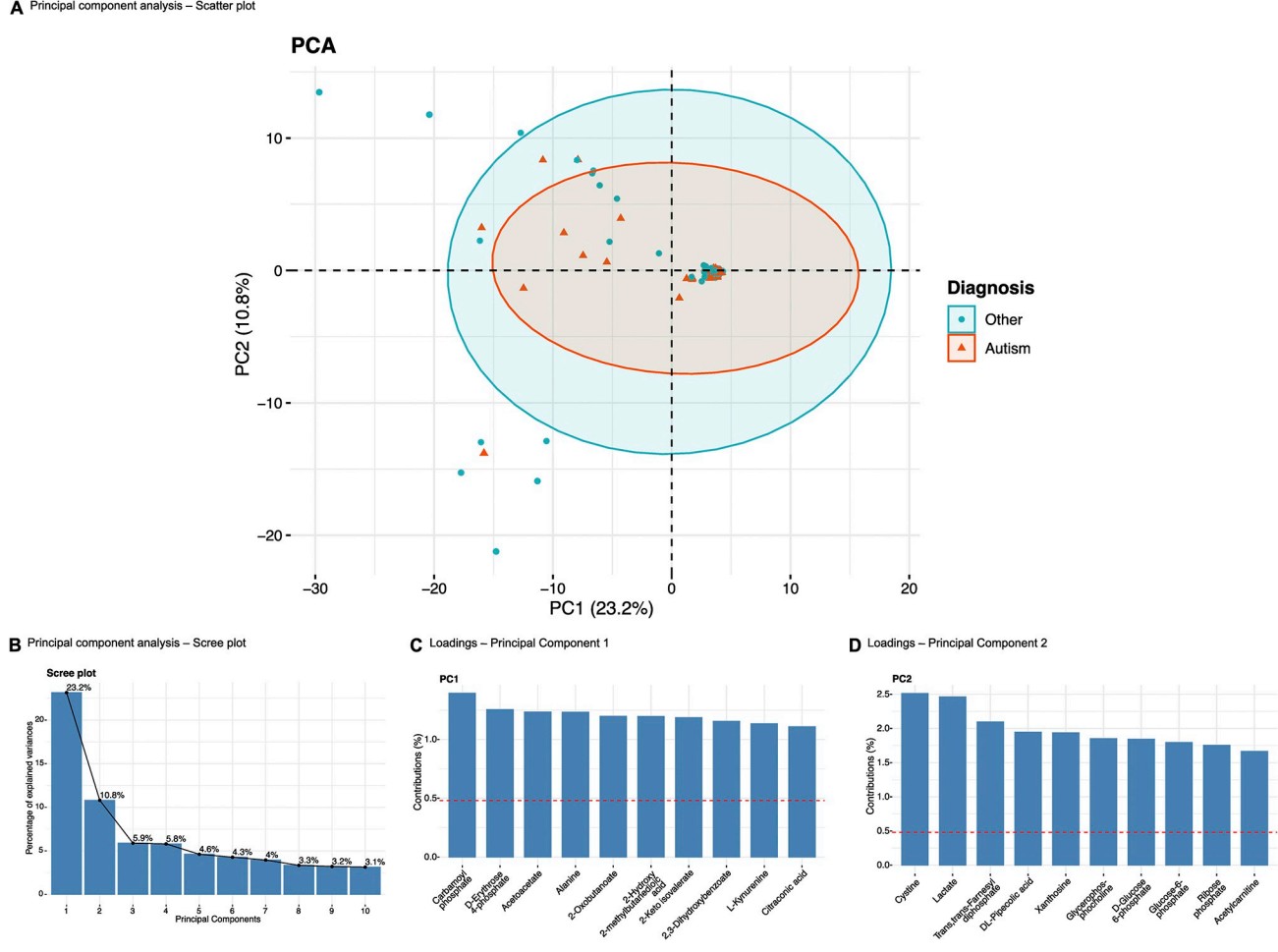

**Fig 1. Principal Component Analysis (PCA) of study cohort.** (A) Scatter plot for individuals based on metabolite contributions and grouped as per autism diagnosis status. (B) Scree plot of eigenvalues of principal components from A. (C) Loading plot of metabolites contributing to principal component 1 (PC1). (D) Loading plot of metabolites contributing to principal component 2 (PC2).

**Table 1. Demographic characteristics of study cohort.**

| Characteristics | Autism (n = 38) | Other (n = 58) |
|---|---|---|
| Age, mean years (SD) [range] | 14.21 (3.26) [8–21] | 15.76 (3.14) [9–23] |
| Sex, n females (%) | 11 (29.95%) | 22 (37.93%) |
| MZ:DZ concordant pairs* | 2:1 | 11:3 |
| Autism discordant pairs (n)* | 14 | |
| MZ | 5 | |
| DZ | 9 | |
| BMI, mean (SD) [range] | 21 (5.41) [13–38] | 21 (3.05) [15–30] |
| Other NDC diagnosis n (%) | 24 (63.16%) | 8 (13.79%) |
| Medication n (%) | 24 (63.16%) | 15 (23.86%) |
| SRS-2, mean (SD) [range] | 74 (18.66) [0–105] | 48 (9.60) [35–78] |
| IQ, mean (SD) [range] | 86 (25.13) [0–138] | 102 (13.81) [65–130] |

SD: Standard Deviation, MZ: Monozygotic, DZ: Dizygotic, BMI: Body Mass Index, NDC: Neurodevelopmental Condition, SRS-2: Social Responsiveness Scale 2nd Edition, IQ: Intelligence Quotient, * Complete twin pairs.

identify a metabolite that was a major driver of the observed effects, with the top 10 loadings having a contribution of ~1.25% to ~1.5% each in PC1 and ~1.75 to ~2.5% each in PC2. There were 9 individuals outside the 95% CI ellipses (Fig 1A), who were designated as outliers and excluded from further analysis, leaving a study cohort with 96 individuals, of which 42 form complete twin pairs. PCA was repeated on this subset of 96 samples to check if the prior removal of outliers better explained the variance of the dataset, however, this was to no avail. More outliers were therefore not removed based on further iterations of the PCA. Moreover, a lack of clear clusters following the PCA (Fig 1A) also ruled out the possibility of using the contemporarily favoured orthogonal partial least squares discriminant analysis (OPLS-DA) model for multivariate analysis in the full cohort [42].

Of these 96 individuals selected for further analysis, the autistic participants had a mean age of 14 years, 11 being female and 24 having a clinical diagnosis of an NDC other than autism. The demographic and clinical features of the study cohort used for further analyses are summarised in Table 1. Specific information on race/ethnicity, socioeconomic status and educational attainment were not recorded.

## Subtle changes in metabolite status across study cohort

The peak areas of each identified metabolite from participants of the study cohort (N = 96) were analysed for association with autism diagnosis using a GEE model, first for the whole cohort (Model A) and then within twin pairs (Model B). We also adjusted the models for the diagnosis status of other NDCs. As expected, from the prior PCA, no metabolite had a significant association with autism after FDR correction for 208 tests. Based on nominal significance, a handful of metabolites were changed (S1A Table). For instance, deoxycholic acid (p = 0.048), orotate (p = 0.042), phenylpyruvate (p = 0.019) and taurine (p = 0.032) were elevated, while carnitine (p = 0.047) was reduced in autism (Fig 2A and S1A Table).

When analysing for differences within twin pairs using the GEE model (Model B), i.e. which corrects for all the shared variation between the twins such as genetics and shared environment, eight metabolites were significantly modulated in the autism group based on the nominal p value (Fig 2A and S1A Table). These were increased hypoxanthine (p = 0.019), diiodothyronine (p = 0.049) and kynurenic acid (p = 0.003), and decreased guanidoacetic acid (p = 0.001), indole (p = 0.001) and L-arginosuccinate (p = 0.031).

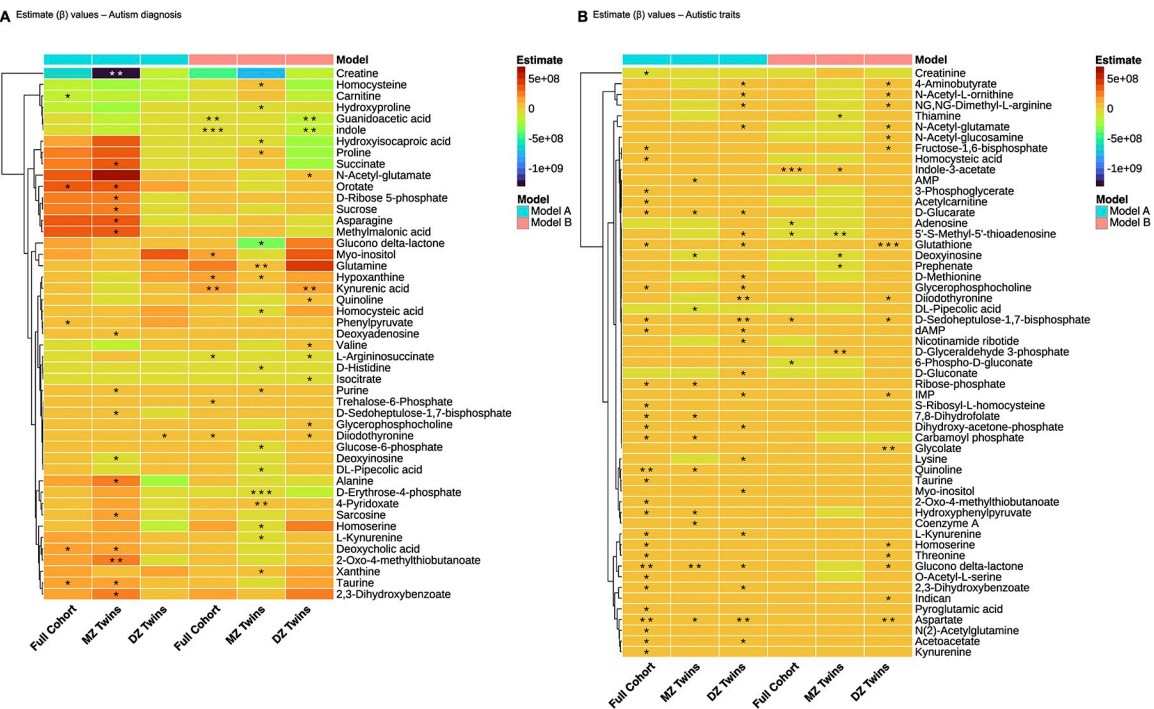

**Fig 2. Heatmap of generalised estimating equations (GEE) model estimates.** (A) Autism diagnosis across full cohort (N = 96), monozygotic twins (n = 52) and dizygotic twins (n = 44) tested using GEE Models A and B. (B) Autistic traits across full cohort, monozygotic twins and dizygotic twins tested using GEE Models A and B. (* p<0.05, ** p<0.01, *** p<0.001).

The study cohort was then subsetted as per zygosity status and the same GEE models were applied for analysis. In the subset of MZ twins (n = 52), 17 metabolites were identified to be significantly different between autistic and other twins (Model A, Fig 2A and S1B Table), including an increase in unique metabolites such as alanine (p = 0.029), asparagine (p = 0.025) and 2-oxo-4-methylthiobutanoate (p = 0.007), and a decrease in creatine (p = 0.001) and deoxyinosine (p = 0.020). When testing for differences within the MZ twin pairs (Model B), 17 metabolites were differentially modulated (Fig 2A and S1B Table), including additional metabolites that were elevated like glutamine (p = 0.008), proline (p = 0.022) and homocysteine (p = 0.037). A decrease was detected in several metabolites, such as glucose-6-phosphate (p = 0.032), L-kynurenine (p = 0.043) and D-histidine (p = 0.014), among others.

In the subset for dizygotic (DZ) twins (n = 44), peak areas of only diidothyronine (p = 0.039) were elevated (Model A, Fig 2A and S1C Table). After accounting for differences within twin pairs in the GEE model (Model B), ten metabolites were significantly modulated (Fig 2A and S1C Table), such as an increase in valine (p = 0.018), quinoline (p = 0.034) and N-acetyl-glutamate (p = 0.046). Reductions were detected in four metabolites including isocitrate (p = 0.018) and indole (p = 0.009).

## Association between autistic traits and metabolite differences

We also investigated associations between the metabolites and quantitative autistic traits measured by the SRS-2 total raw scores [27–29]. We analysed the association between the peak areas of each metabolite identified in the study cohort (N = 96) with the SRS-2 using the GEE models and adjusted for IQ (Wechsler GAI) [30]. Again, no metabolite reached statistical significance after FDR multiple test correction. From Model A, 30 nominally significant

metabolites were detected (Fig 2B and S2A Table). Amongst those elevated were acetylcarnitine (p = 0.023), aspartate (p = 0.002) and hydroxyphenylpyruvate (p = 0.016); and the only metabolite to be reduced was creatine (p = 0.027). When testing for differences within the twin pairs (Model B), five metabolites crossed the nominal significance threshold (Fig 2B and S2A Table), including a decrease in adenosine (p = 0.025) and 6-phospho-d-gluconate (p = 0.033), and an increase in indole-3-acetate (p = 0.0001, adjusted p = 0.031).

We further explored the association in the zygosity groups for the autistic traits and the metabolites. In the MZ subset (n = 52), 12 metabolites demonstrated nominally significant changes when applying the GEE model (Model A, Fig 2B and S2B Table). An elevation was detected in coenzyme A (p = 0.044), AMP (p = 0.041) and a decline in levels of DL-pipecolic acid (p = 0.024). Following testing for differences within the twin pairs (Model B), six metabolites were identified (Fig 2B and S2B Table), which included an increase in prephenate (p = 0.027) and D-glycerylaldehyde 3-phosphate (p = 0.003) and a decrease in 5'-S-Methyl-5'-thioadenosine (p = 0.001) and thiamine (p = 0.032).

In the DZ subset (n = 44), 23 metabolites were significantly elevated (p<0.05) in association with the autistic traits (Model A, Fig 2B and S2C Table). These were, to name a few, D-methionine (p = 0.047), D-gluconate (p = 0.038), myo-inositol (p = 0.047) and lysine (p = 0.043). When accounting for differences between the twin pairs (Model B), 16 metabolites demonstrated significant changes (Fig 2B and S2C Table) where none were reduced, including glycolate (p = 0.009), 4-aminobutyrate (p = 0.035) and IMP (p = 0.012).

## Enrichment of identified metabolites in biochemical pathways

The nominally significant metabolites identified from the GEE models were used for ORA as described in the methods. Ten biochemical pathways were identified to be enriched in the autistic twins based on the full cohort analysis (N = 96, Model A) (Fig 3A and 3B), out of which there was significant enrichment only in the bile acid biosynthesis pathway (p = 0.035, S3A Table). When ORA analysis was done for the metabolites found to be nominally significant for autism diagnosis between the twin pairs (Model B), ten pathways were identified (Fig 3C and 3D), where only the arginine and proline metabolism was significant (p = 0.024, S3B Table).

For the models based on autistic traits in the full cohort (N = 96, Model A), 43 biochemical pathways were enriched (Fig 4B), out of which two crossed the significance threshold (S3C Table): glutathione metabolism (p = 0.0503) and glutamate metabolism (p = 0.0506). The enrichment dot-plot of the top ten pathways is depicted in Fig 4A. On the other hand, eight biochemical pathways were enriched from significant metabolites identified after testing for autistic traits between the twin pairs (Model B, S3D Table), of which methionine metabolism (p = 0.001) was significantly detected (Fig 4C and 4D).

## Discussion

In this study, we explored the urine metabolome of a subset of individuals from the RATSS cohort [19, 20] using untargeted mass-spectrometry. While only a single metabolite reached significance after FDR correction, we identified several metabolites as nominally significant for autism diagnosis and autistic traits. Furthermore, the identified metabolites were enriched in a few relevant biochemical pathways. Increased detection of significant metabolites was found for the autistic traits, rather than for the autism diagnoses.

Our study is the first to evaluate differences in the urine metabolome of an autism twin cohort and utilise information on co-occurring NDCs in our analyses. This is an important aspect to account for, when investigating suitable biomarkers in lieu of the high prevalence of

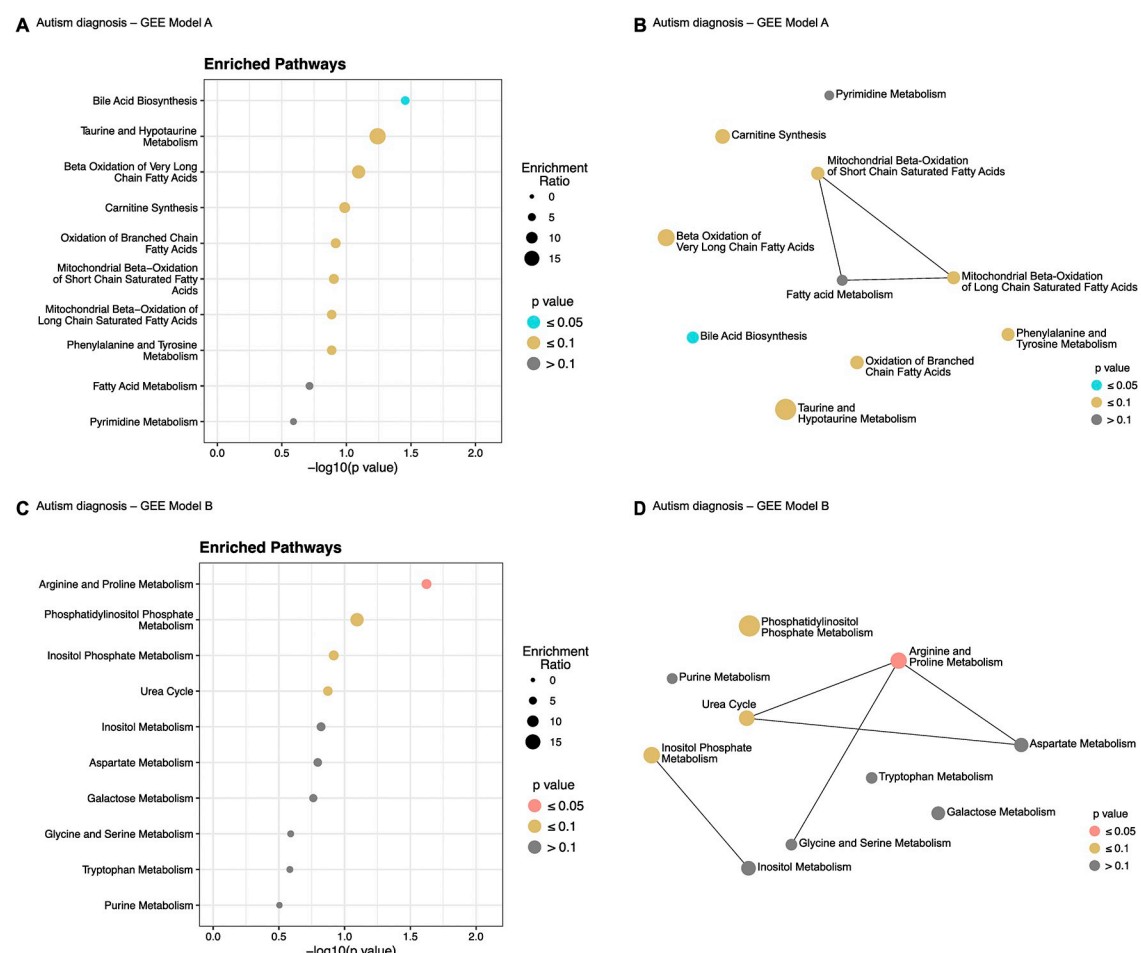

**Fig 3. Pathway enrichment of significant metabolites based on autism diagnosis.** (A) Enrichment dot-plot for top 10 enriched pathways from GEE Model A. (B) Interaction network for enriched pathways from GEE Model A. (C) Enrichment dot-plot for top 10 enriched pathways from GEE Model B. (D) Interaction network for enriched pathways from GEE Model B.

co-occurring NDCs in autism [43]. The size of our study cohort (N = 96, with n = 38 diagnosed with autism) is larger when compared to several recently published studies using urine-based mass spectrometry, such as 60 available samples from an otherwise larger cohort [44], 57 participants [13] and 14 autism discordant sibling pairs [15]. The twin cohort also gives us the possibility to analyse the effects of genetic and environmental factors on these potential biomarkers. Despite this, we were not able to find any metabolites that reached the multiple test correction significance. However, a few were nominally associated similarly in the whole cohort and within twin pair analyses (Fig 2A). Our results show that the contribution of the shared factors, including genetics, in the metabolite profiles is strong and therefore it is difficult to find robust biomarkers of autism.

Our study is also the first to use autistic traits as measured by the SRS-2 as a predictor of metabolite changes. Predominantly, we found a large number of metabolites being associated with autistic traits rather than autism diagnosis. This shows potential that in the future, metabolite profiles could be used for detecting differences in continuous autism operationalisations. We observed a significant positive association between indole-3-acetate and autistic traits within the twin pairs. Elevated levels have been previously detected in autistic individuals

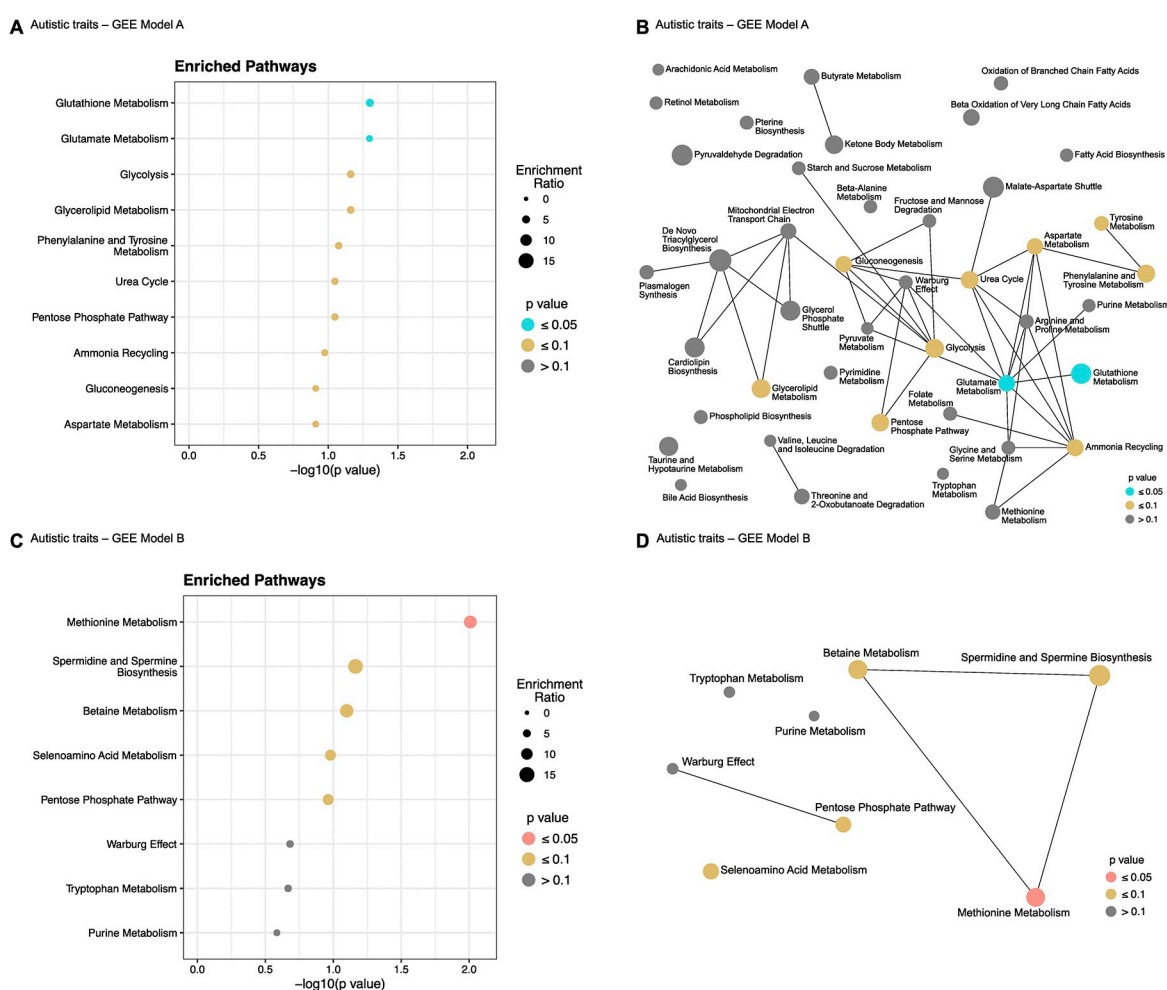

**Fig 4. Pathway enrichment of significant metabolites based on autistic traits.** (A) Enrichment dot-plot for top 10 enriched pathways from GEE Model A. (B) Interaction network for enriched pathways from GEE Model A. (C) Enrichment dot-plot for top 10 enriched pathways from GEE Model B. (D) Interaction network for enriched pathways from GEE Model B.

[13, 45]. However, a decline has also been reported [46]. Indole-3-acetate is a known by-product of the tryptophan metabolism modulated by the gut microbiome [47, 48], with growing evidence indicating a dysbiosis [49, 50].

Despite the fact that we only found nominally associated metabolites, our findings align with previously reported urine-based metabolomic studies for autism diagnosis. For instance, we observed elevated levels of phenylpyruvate [14, 15] and taurine [11, 12, 51, 52], as noted earlier in several studies. We also show decreased levels of carnitine, similarly to what was reported in other biological samples [53, 54]. However, direct comparisons with metabolomic studies using different biological samples should be critically evaluated due to lack of information about any correlations between such reports, as observed in other multifactorial pathologies [55]. Our observations are unique when it comes to the association with autistic traits, as these have not been used as a predictor in metabolomic studies of autism before. While individual metabolites provide a snapshot into specific altered metabolic processes, interpretation of pathway level changes may be more relevant in multifactorial conditions like autism [56].

Several of the enriched pathways that were detected in this study are related to energy metabolism and mitochondrial function, which are crucial in the developing brain [57]. There

are earlier studies that have linked such metabolic changes with autism physiology. These include the genetics [58] and functional differences [59] that contribute to atypical mitochondrial function in autism [60, 61]. We have also identified pathways related to amino acid metabolism, although with relatively lower confidence. This may nevertheless still prove to be an interesting aspect to investigate further, particularly with the recently growing evidence in this direction [62–64].

Even with a relatively larger cohort, our study like several others is limited by the size of the clinical cohort to identify robust and reproducible urine-based biomarkers of autism, if there are any. Despite, collecting the urine samples in the same manner for all study participants, variations in the sample handling could affect the metabolite profiles. Furthermore, since urine is an excretory by-product, dietary choices can influence the urine metabolome [65], both in neurotypical and autistic individuals. Future studies should aim to include this information in their analyses, accompanied with knowledge about the gut-microbiome status of study participants, as this can also be potentially associated with autism status [66]. It would also be essential to account for puberty related changes in the metabolism of adolescents while characterising the metabolome [67, 68]. Lastly, there is a need to replicate such studies in several international cohorts to test the value of identified biomarkers in different populations.

In conclusion, we have identified several urinary metabolites that show nominal association with autism diagnosis status and autistic traits. We have also determined the enrichment of the significant features in metabolic pathways. While our findings highlight no robust urine-based biomarkers for autism, this ever-increasing knowledgebase can serve as signposts for investigations in the future.

## Supporting information

**S1 Table. GEE model for metabolites and autism diagnosis.** (A) Outcomes from modelling in full cohort. (B) Outcomes from modelling in monozygotic twins subsetted from full cohort. (C) Outcomes from modelling in dizygotic twins subsetted from full cohort.
(XLSX)

**S2 Table. GEE model for metabolites and autistic traits.** (A) Outcomes from modelling in full cohort. (B) Outcomes from modelling in monozygotic twins subsetted from full cohort. (C) Outcomes from modelling in dizygotic twins subsetted from full cohort.
(XLSX)

**S3 Table. Over representation analysis of significant metabolites.** (A) Enriched pathways from autism diagnosis and GEE Model A. (B) Enriched pathways from autism diagnosis and GEE Model B. (C) Enriched pathways from autistic traits and GEE Model A. (D) Enriched pathways from autistic traits and GEE Model B.
(XLSX)

## Acknowledgments

We would like to thank all twins and parents who have participated in this research. We would also like to thank the RATSS team, especially Anna Pilfalk, Dr. Karl Lundin Remnelius, Dr. Johan Isaksson and Dr. Janina Neufeld for their valuable contribution to the work presented in this study. Additionally, we would like thank Prof. Antonio Persico for early discussion on designing the project. We acknowledge the Swedish Twin Registry for data access and urine analysis by the Proteomics and Metabolomics Facility, Department of Ecological and Biological Sciences, University of Tuscia, Viterbo, Italy and Prof. Lello Zolla.

## Author Contributions

**Conceptualization:** Abishek Arora, Sven Bölte, Kristiina Tammimies.

**Data curation:** Abishek Arora, Kristiina Tammimies.

**Formal analysis:** Abishek Arora.

**Funding acquisition:** Abishek Arora, Sven Bölte, Kristiina Tammimies.

**Investigation:** Abishek Arora, Sven Bölte, Kristiina Tammimies.

**Methodology:** Abishek Arora, Francesca Mastropasqua, Kristiina Tammimies.

**Project administration:** Abishek Arora, Sven Bölte, Kristiina Tammimies.

**Resources:** Abishek Arora, Sven Bölte, Kristiina Tammimies.

**Software:** Abishek Arora.

**Supervision:** Sven Bölte, Kristiina Tammimies.

**Validation:** Abishek Arora.

**Visualization:** Abishek Arora, Kristiina Tammimies.

**Writing – original draft:** Abishek Arora, Francesca Mastropasqua, Kristiina Tammimies.

**Writing – review & editing:** Abishek Arora, Francesca Mastropasqua, Sven Bölte, Kristiina Tammimies.

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
