## [Decision Letter · Decision Letter 0]

15 Apr 2024

PONE-D-23-39743Urine metabolomic profiles of autism and autistic traits – a twin studyPLOS ONE

Dear Dr. Tammimies,

Thank you for submitting your manuscript to PLOS ONE. After careful consideration, we feel that it has merit but does not fully meet PLOS ONE’s publication criteria as it currently stands. Therefore, we invite you to submit a revised version of the manuscript that addresses the points raised during the review process.

We look forward to receiving your revised manuscript.

Kind regards,

Claudia Brogna

Academic Editor

PLOS ONE

3. Thank you for stating the following financial disclosure: "The project was supported by the Swedish Research Council (S.B., and K.T.), Swedish

Foundation for Strategic Research (K.T.), the Swedish Brain Foundation – Hjärnfonden (K.T.), the Harald and Greta Jeanssons Foundations (K.T.), Åke Wiberg Foundation (K.T.), Strategic Research Area Neuroscience Stratneuro (K.T.), The Swedish Foundation for International Cooperation in Research and Higher Education STINT (K.T.), and Board of Research at Karolinska Institutet (K.T.). " 

4. In the online submission form, you indicated that the mass spectrometry data is accessible from the corresponding author after necessary

clearances. The utilised code is available on GitHub (https://github.com/Tammimies-

Lab/RATSS-Metabolomics) or available upon request from the corresponding author.. 

Reviewers' comments:

Reviewer's Responses to Questions

**Comments to the Author**

1. Is the manuscript technically sound, and do the data support the conclusions?

Reviewer #1: Yes

2. Has the statistical analysis been performed appropriately and rigorously? 

Reviewer #1: No

3. Have the authors made all data underlying the findings in their manuscript fully available?

Reviewer #1: Yes

4. Is the manuscript presented in an intelligible fashion and written in standard English?

Reviewer #1: Yes

5. Review Comments to the Author

Reviewer #1: Autism disorders were well-recognized as a metabolic disease in the past decades. Urinary biomarkers are very useful to identify this spectrum. So it needs more broad consideration to get the conclusion with negative results.

6. PLOS authors have the option to publish the peer review history of their article (what does this mean?). If published, this will include your full peer review and any attached files.

Reviewer #1: No

---

## [Author Response · Author response to Decision Letter 0]

6 Jun 2024

Reviewer

The metabolomic analysis in twin-cohort to discover urinary biomarker of autism is interesting and of significance. It is valuable to tease out many confounding factors, as gene and environment factors. Although no significant metabolic drivers for autism diagnosis were detected when controlling for other neurodevelopmental conditions. Some nominally changes were found within the twin pairs. 

Nevertheless, we have the following questions：

1. For PCA analysis, 9 individuals were outliers and excluded, but no better clustering and explained in further repeated PCA. What is the conceptual basis and meaning for excluding individual?

As rightly stated, following PCA, 9 individuals were designated as outliers and excluded as they were outside the 95% confidence interval ellipses of the scatter plot for the most explanato-ry principal components (PC1 and PC2, Figure 1A). Further repeated PCAs did not improve clustering and were therefore not included. This rationale and our approach to handling outliers in this study is stated in the manuscript (lines 208 – 213). The removal of the outliers was done based on recommendations in the field (PMID: 38179935). Nevertheless, only 8.57% of the total study participants (N=105) were excluded, which would not have a statistical impact on our findings.

2. For untargeted metabolomics, features identified by the UPLC-MS should be listed and offer more details, as m/z, rt or fragmentation similarity, etc. It is important for ex-hibiting confidence in annotating metabolites.

The 208 features identified following untargeted metabolomic using UHPLC-MS have been stated in Table S1 and Table S2. The UHPLC-MS analysis and annotation were performed by the Proteomics and Metabolomics Facility of the University of Tuscia, Italy, using their stand-ard established pipeline (lines 114 – 148). This analytical approach has also been utilised in previously published papers (PMIDs 32512190 and 27904735) by members of the core facili-ty. Unfortunately, we were not able to get the m/z, rt or fragmentation similarity from the core facility but only the standard annotated metabolites as reported. As the metabolomics analysis was performed on a clinical cohort, further data cannot be included in this manuscript in lieu of ethical considerations.

3. Urine is an unstable milieu, collection time and method may affect the results. While no information was given in the manuscript. Please explain your consideration of these factors.

Thank you for noting, we have now included more information about how the urine was col-lected from the participants (lines 110 – 116). We additionally added a sentence about the col-lection of the urine samples and instability to the manuscript’s discussion (lines: 381 – 382). 

4. As mentioned in the manuscript, indole-3-acetate is positive associated with autistic traits within the twin pairs. If it is possible to assess the potential of indole-3-acetate as a predictor?

Although, indole-3-acetate is positively associated with autistic traits within the twin-pairs, it would not be appropriate to access its potential role as a predictor due to the confounding ef-fects of participant genetics.

Academic Editor

The manuscript has been updated to meet the shared style requirements of PLOS ONE, includ-ing the naming of all uploaded files.

We have ensured that the grant information provided at the time of resubmission matches that which is included in the manuscript (lines 415 – 422).

3. Thank you for stating the following financial disclosure: "The project was supported by the Swedish Research Council (S.B., and K.T.), Swedish Foundation for Strategic Research (K.T.), the Swedish Brain Foundation – Hjärnfonden (K.T.), the Harald and Greta Jeanssons Foundations (K.T.), Åke Wiberg Foundation (K.T.), Strategic Research Area Neuroscience Stratneuro (K.T.), The Swedish Foundation for International Coop-eration in Research and Higher Education STINT (K.T.), and Board of Research at Ka-rolinska Institutet (K.T.). " Please state what role the funders took in the study. If the funders had no role, please state: ""The funders had no role in study design, data col-lection and analysis, decision to publish, or preparation of the manuscript." 

If this statement is not correct you must amend it as needed. Please include this amend-ed Role of Funder statement in your cover letter; we will change the online submission form on your behalf.

The funders had no role in this study and this has been stated in the funding information (lines 420 – 422), as shared in the above-mentioned statement.

4. In the online submission form, you indicated that the mass spectrometry data is ac-cessible from the corresponding author after necessary clearances. The utilised code is available on GitHub (https://github.com/Tammimies-Lab/RATSS-Metabolomics) or available upon request from the corresponding author.

All PLOS journals now require all data underlying the findings described in their man-uscript to be freely available to other researchers, either 1. In a public repository, 2. Within the manuscript itself, or 3. Uploaded as supplementary information.

This policy applies to all data except where public deposition would breach compliance with the protocol approved by your research ethics board. If your data cannot be made publicly available for ethical or legal reasons (e.g., public availability would compromise patient privacy), please explain your reasons on resubmission and your exemption re-quest will be escalated for approval. 

The data included in this manuscript pertains to clinical research and cannot be made publicly available for ethical and legal reasons as this would compromise patient privacy.

The ethics statement is included in the Materials and methods section (lines 86 – 88) and has been removed from any other section of the manuscript.

6. Please include captions for your Supporting Information files at the end of your manuscript, and update any in-text citations to match accordingly. Please see our Sup-porting Information guidelines for more information: http://journals.plos.org/plosone/s/supporting-information.

Captions for the Supporting information files have been included in the manuscript (lines 437 – 448) and match the in-text citations.

---

## [Editor Report · Decision Letter 1]

19 Jul 2024

Urine metabolomic profiles of autism and autistic traits – a twin study

PONE-D-23-39743R1

Dear Dr.Kristiina Tammimies,

We’re pleased to inform you that your manuscript has been judged scientifically suitable for publication and will be formally accepted for publication once it meets all outstanding technical requirements.

Kind regards,

Claudia Brogna

Academic Editor

PLOS ONE

---

## [Editor Report · Acceptance letter]

24 Jul 2024

PONE-D-23-39743R1 

PLOS ONE

Dear Dr. Tammimies, 

I'm pleased to inform you that your manuscript has been deemed suitable for publication in PLOS ONE. Congratulations! Your manuscript is now being handed over to our production team.

Kind regards, 

on behalf of

Dr. Claudia Brogna 

Academic Editor

PLOS ONE